

# Traditional art design expression based on embedded system development

Yan Cui[1,2] and Amer Shakir Bin Zainol[2]

[1] Academy of Arts, Zhengzhou Business University, Zhengzhou, Henan, China
[2] College of Creative Arts, University Technology MARA (UITM), Shah Alam, Selangor, Malaysia

## ABSTRACT

This article describes constructing an embedded system for a painting art and style presentation platform, achieving the automatic integration of digital painting art with traditional art design. The frontend components are designed using the Bootstrap framework, with Django as the web development framework and TensorFlow architecture integrated into the code. Furthermore, the Inception module and residual connections are introduced to optimize the visual geometry group (VGG) network for recognizing and analyzing image texture features. Compared to other models, experimental results indicate that the proposed model demonstrates a 2.6% increase in image style classification accuracy, reaching 87.34% and 95.33% in architectural and landscape image classification, respectively. The system's operational outcomes reveal that the proposed platform alleviates the burden on the logical function modules of the system, enhances scalability, and promotes the automated fusion of digital painting art with traditional art design expression.

# INTRODUCTION

Digital painting art has emerged as a novel artistic form driven by the ongoing evolution of contemporary science (*Sochorová & Jamriška, 2021*). Unlike conventional painting methods, digital art utilizes tools such as keyboards, mice, drawing tablets, displays, and peripherals to create a digital canvas. It allows for effortless manipulation, refinement, duplication, and widespread dissemination of intricate digital imagery. On the other hand, traditional art design is rooted in hand drawing, supplemented by various tools. It relies on art painting skills and emphasizes cultivating fundamental abilities and artistic accomplishments, embodying historical tradition and broad cultural charm (*Han & Li, 2023*). In the information age, digital painting art has influenced traditional art's observation and design modes, leading to conflicts in expression modes (*Zhao & Lee, 2018*). Digital painting art, as a new artistic method, involves virtual painting art realized through computers and various drawing software. Its speed and effectiveness have rapidly integrated into people's lives (*Li, 2023*). Digital painting is an inclusive art form capable of imitating traditional art while possessing its digital language, making it a vibrant art form. The harmonious fusion of digital painting art and traditional art design requires a profound amalgamation of traditional artistic principles and digital painting elements. It

Corresponding author
Amer Shakir Bin Zainol,
17516225170@163.com

fosters the creation of distinctive artistic works that contribute to both art forms' mutual growth and advancement (*Ye, Wang & Wang, 2021*). The fusion of image processing with an embedded platform ensures the portability of this synthesis between digital painting art and traditional art design, diversifies embedded systems and bestows practical significance and value upon this endeavor. Unlike digital painting, traditional painting involves physical materials such as pigments, brushes, and other tools to create art on paper or cloth (*Ma et al., 2020*). Despite their differences, these two forms of art are closely related, with digital painting art serving as a supplement to traditional art. In digital painting art, we can discern the influence and concepts of traditional art; conversely, digital painting art impacts traditional art design (*Lan, 2018*). As the field of painting continues to evolve, it gives rise to an increasing number of exceptional artworks. Proficiency in external aspects, such as technical skills, manifests theoretical acumen and complements the artistic process.

Consequently, the symbiotic fusion of digital painting art and traditional art design holds profound practical importance in the progression of art design. In the ongoing convergence of digital painting art and traditional art, style transfer based on convolutional neural networks (CNN) has emerged as the predominant approach. This method leverages convolutional kernels to perform convolution operations and replicate stylistic attributes from one image onto a target image. Each layer within the CNN holds a wealth of visual feature data. The initial layers retain most of the input image's low-level characteristics, including shape, position, color, and texture. As the CNN deepens, the feature information extracted by the convolutional kernel becomes increasingly semantic. The CNN extracts visual information from the input image and attempts to transform it into the desired semantic classification domain.

However, implementing CNN-based systems and platforms poses formidable challenges that demand substantial computational resources. The seamless integration of digital and traditional painting art must more readily facilitate these challenges. To effectively and continuously integrate digital painting art and traditional art across diverse platforms, a strategic approach involves reengineering the existing CNN-based platform, and this entails a meticulous design of the platform's various modules, with a concerted effort to diminish inter-component coupling and correlations. Notably, when deploying a platform for rendering painting styles, one must contend with elevated costs, particularly for dedicated GPUs on cloud servers. It is imperative to strike a balance to mitigate expenses by ensuring the system's hardware configuration remains within reasonable bounds. To address these issues, we propose the integration of digital painting art and traditional art design expression through an embedded system. The main contributions are as follows.

Leveraging an embedded system, we have successfully constructed a rendering platform dedicated to painting art styles, enabling the automatic integration of digital and traditional art. Within the network layer, each node's data encapsulates unique, context-specific information, showcasing variations in depth and surface characteristics across the network.

The designed rendering platform for painting art styles facilitates the automatic switching and style conversion between digital painting art and traditional art design. Designers input paintings of the same style into the computer to complete a figure painting. The system then autonomously classifies, calculates, and analyzes them through the artificial neural

network. Subsequently, based on the learned visual style, these images are transformed into digital paintings, thereby realizing the integration and innovation of digital painting art and traditional art design.

## RELATED WORK

Conventional image style transfer algorithms have primarily relied on collecting textures from style maps and applying them to content maps (*Li & Chen, 2023*). However, these algorithms extract mainly low-level image features, resulting in suboptimal synthesis outcomes, especially when dealing with images featuring intricate colors and textures (*Kong et al., 2023*). They often need faster processing speeds, making them impractical for real-world scenarios.

Nevertheless, the continuous advancements in artificial intelligence technology have opened new avenues, allowing for the achievement of artistic image stylization through convolutional neural networks (CNN) (*Gupta et al., 2022*). Although these algorithms have shown remarkable performance after extensive refinement, they still face several challenges. Firstly, there's an ongoing quest to make reasonable parameter adjustments. Secondly, streamlining pre-trained models remains a challenge.

In 2006, *Winnemöller, Olsen & Gooch (2006)* pioneered the concept of an image-based cartoon-stylized rendering system, achieving real-time image cartoon rendering for the first time. While image filtering technology is more mature in theory, resulting in faster and more stable algorithm implementations, its capability to realize diverse rendering styles remains significantly constrained. In 2015, *Gatys, Ecker & Bethge (2015)* introduced an image style transfer algorithm based on CNN, which effectively disentangles and reconstructs the content and style of any image using deep neural networks. This research concept has provided a fundamental reference direction for the evolution of image style transfer technology.

Furthermore, the traditional Image Quilting algorithm fails to account for the modifications required in the target image's direction field characteristics during patch stitching. To address these limitations, *Shen, Tang & Xu (2023)* introduced an enhanced Image Quilting image style transfer algorithm, which relies on guiding the target image's structure tensor's direction to ensure the rational transmission of directional information onto the target image. *Li et al. (2019)* proposed a linear feed-forward network based on statistics to achieve a high-quality arbitrary style transfer algorithm using image feature extraction by the visual geometry group (VGG) network. This method can adapt well to the conversion between arbitrary reference styles and content images and has also yielded positive results in video style transfer. *Deng et al. (2020)* proposed a multi-adaptive arbitrary style transfer network to better adapt to artistic styles. This approach employs two self-adaptive modules and one co-adaptive module to adjust the style distribution of reference images to match content features, achieving favorable results across various styles. *Zhao et al. (2020)* devised a framework for portrait-style transfer using an enhanced deep convolutional neural network (DCNN). Initially, this framework automatically partitions the style portrait and content portrait into seven segments, allowing for the capture of

style attributes from the elements within the style image while preserving the overarching structure of the content portrait. Then, the mask of these seven parts is added to the trained DCNN to enhance the feature map in some specific layers of the depth model, facilitating the artistic style transfer of the portrait (*Zhao et al., 2020*).

Despite its capacity to produce impressive stylized images, image style transfer based on image optimization still needs to grapple with computational inefficiencies. An alternative approach, model optimization for style transfer, addresses this issue by utilizing a trained network generation model capable of swiftly synthesizing stylized results; this is achieved by optimizing the feed-forward neural network using various graphs for one or more style images. However, the prevailing single-model, single-style paradigm requires training a distinct generative neural network for each specific style image, a time-consuming and inflexible process. For instance, many paintings, such as Impressionist artworks, share similar painting strokes but exhibit different color palettes, making using separate neural networks for each style redundant.

To overcome this limitation, *Tao (2022)* proposes matching style blocks with content blocks in the feature space of a pre-trained VGG network, facilitating the exchange of content and style blocks. This exchange results in a feature map, which is then promptly reconstructed using an image reconstruction algorithm. While this algorithm offers greater flexibility and can generate diverse style images, the stylized outcomes sometimes must be revised. Content blocks are occasionally swapped with style blocks that inadequately represent the desired style during the exchange, resulting in subpar style representation.

Additionally, *Nizan & Tal (2020)* developed a Council-GAN model, which breaks the constraint of cyclic consistency and employs cooperation among multiple generators to address the typical defects of GAN and eliminate traces unrelated to the input image in the generated image. This model has achieved positive results in various image manipulation tasks, including gender transition, self-portraits, two-dimensional animation, and glass removal. However, the collaboration between its multiple generators and discriminators increases the time and space complexity of network training (*Nizan & Tal, 2020*).

*Cui et al. (2021)* propose bilateral convolutional blocks and feature fusion strategies to achieve visual smoothness and introduce self-supervised semantic networks to predict semantic information, maintaining the semantic integrity of content images. This approach addresses common problems, such as missing content and blurriness, in neural style transfer (NST). While it can convert portrait photos into realistic and cartoon-style images, it sometimes experiences local information loss (*Cui et al., 2021*).

Despite the simplicity of the network structure in the mentioned method, the complexity lies in its intricate modeling, requiring repeated iterations of the network training model. This hampers application efficiency. Furthermore, the absence of GPU acceleration leads to time-consuming computations, and the goal of real-time application still needs to be discovered. Consequently, the transfer effect tends to be implausible, resulting in deformed and distorted textures in the generated images.

## MATERIAL AND METHOD

### Data source and pre-processing

We crawled the internet and selected 300 natural images and 300 artistic image-style paintings as the data materials for this experiment. Since the chosen image dataset is relatively small, we adopted data augmentation technology in this experiment to increase the size of both image datasets by a factor of three, respectively. Before training the model, the images underwent pre-processing. Firstly, we reduced the resolution of all images to $256 \times 256$. For the image data with different channels, we standardized the mean and variance of the channels to make the data distribution in each channel more consistent. Then, we normalized the image data to compress the pixel values within the range of (0,1), which helped expedite the model training and optimization process. Furthermore, we employed an image-denoising technique, precisely Gaussian filtering, to mitigate noise interference within the images.

The basic formula for Gaussian filtering is expressed as follows:

$$G(x, y) = \frac{1}{2\pi\sigma^2} \exp\left(-\frac{x^2 + y^2}{2\sigma^2}\right) * I(x, y)$$

where:
- $G(x, y)$ is the image after Gaussian filtering.
- $I(x, y)$ is the original image.
- $\sigma$ is the standard deviation of the Gaussian kernel.
- $(x, y)$ represents the pixel coordinates in the image.
- $*$ denotes the convolution operation.

This approach augments the diversity of the training dataset and enhances the model's generalization capacity through a sequence of random transformations, including random rotation, translation, scaling, flipping, and so forth. By incorporating these pre-processing steps, the model can more effectively discern image features, improving overall model accuracy.

### Hardware planning

An embedded system, or an embedded computer system, incorporates computer technology and integrates numerous integrated circuits. An embedded hardware platform is a computer system designed around large-scale integrated circuits (ICs) and includes various external devices and processors (*Cheng, 2018*; *Zhuo, 2016*).

The embedded operating system serves as the core of embedded products. Choosing an embedded operating system becomes a critical consideration when designing software, as it directly impacts the development tool's complexity, development cycle duration, and system portability. In line with the platform's functional requirements, the selected operating system should demonstrate superior portability, extensive compatibility with peripheral hardware, abundant shared resources, a wide array of drivers, minimal memory usage, and reduced hardware resource demands. During the development process, Linux has emerged as a preferred operating system for terminals due to its compatibility with source code and APIs. Building upon Linux, we have implemented an embedded operating

system for creating rendering drawing styles. Linux delivers outstanding performance across diverse network configurations and workstations, efficiently accommodating many concurrent users while effectively managing their operations. Furthermore, Linux functions as a multitasking operating system, proficiently executing numerous tasks concurrently without causing system slowdowns, as demonstrated by its ability to handle large file downloads without compromising system performance.

The platform's hardware, centered around ARM architecture, has undergone the entire system transplantation process, including driver and program design. The platform for rendering drawing styles has found extensive applications in various fields, especially in art education and cultural work design, with widespread adoption and a long operational history. Factors such as resistance to interference, stability, cost, and chip-specific technical support are considered when selecting an ARM processor.

## Software design

The design principles for the rendering drawing style platform encompass two pivotal facets: one caters to user-centric considerations, while the other revolves around system development, deployment, and operation. Regarding users, the platform must first adhere to the tenets of stability, user-friendliness, and an exceptional user experience. In this context, stability pertains to the quality of the system's code. Ensuring the reliability and security of the code necessitates robust error management capabilities. Any encountered errors should be promptly reported and addressed to prevent system paralysis.

Moreover, user-centricity extends to the frontend interface, emphasizing the need for clear icons and buttons (*Alsabki, 2022*). Additionally, the platform should provide affirmative feedback for every user operation. A superior user experience entails expeditious image rendering, preventing users from enduring lengthy waiting times; this necessitates minimizing the runtime of background rendering modules, ensuring optimal performance.

The platform for rendering drawing styles follows a B/S architecture, hierarchically structured into three tiers: the presentation, business logic, and data layers. The data layer encompasses the repository of drawing image data stored on the platform. It interacts with the business logic layer, responding to data requests initiated by the latter. The business logic layer is primarily responsible for managing various aspects of the platform's operations, including information servers, WEB servers, and style servers (*Zhao & Lee, 2018*). After executing the requisite business processes, the outcomes are relayed to the presentation or data layers. The presentation layer, represented by the WEB interface, is accessible through the server's published artwork addresses. It manifests the system's logic, showcasing rendered works and digital paintings. The overall planning of the platform is depicted in Fig. 1.

The platform mainly consists of two modules. The first module is the user-facing frontend interface, designed to be simple and visually appealing, capable of interacting with the background and displaying the output results. The second module constitutes the platform's back end.

Within this framework, the business logic function module in the back end assumes a central role in managing diverse requests and orchestrating the process pool. This module

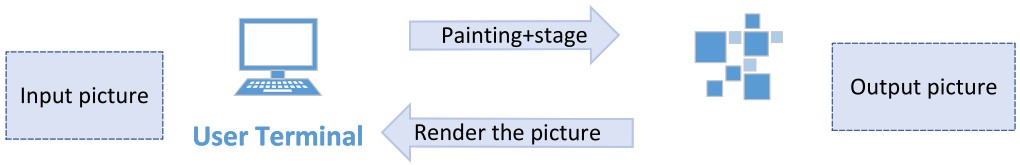

**Figure 1** Overall planning of the platform for rendering drawing style.

carries the dual responsibility of verifying the legitimacy of each request and judiciously dispatching requests to the processing poo land; this ensures that user-initiated requests do not result in prolonged unavailability. The URL processing module plays a crucial role as well. Its primary function is to validate the authenticity of each connection and route the corresponding request to the appropriate module (*Stanisław & Leszek, 2021*).

## Core function design

The rendering module is the central function within the drawing style rendering platform, seamlessly integrated into its background. This critical rendering function module leverages TensorFlow to realize the rendering of diverse painting styles. The presentation module must pre-train a spectrum of styles, enabling users to select accordingly (*Li, 2023*). In the design of the style presentation module, this section mainly introduces some existing types of transplantation networks. Then, it uses TensorFlow to generate the code, which can pre-extract artists' works with different styles.

TensorFlow, an open-source machine learning framework, features specific modules essential for developing and training deep learning models; one such module is tf Keras, an advanced high-level neural network API within TensorFlow. It facilitates constructing and training deep learning models through the tf.Keras.Sequential interface, allowing for the seamless assembly or customization of layered models.

The data module plays a crucial role by providing tools for creating high-performance input pipelines and optimizing data input for machine learning tasks. These modules contribute to TensorFlow's overall efficiency and flexibility, enabling researchers and practitioners to quickly design and implement sophisticated neural network architectures.

The primary network used in the platform for rendering drawing styles is the one proposed by *Gatys, Ecker & Bethge (2015)*. The style transfer network analyzes the image's texture and content features through VGGNet. Examining these features determines two different loss functions: the content loss function and the style loss function (*Ma et al., 2020*). After repeated training, the loss functions are minimized, allowing the network to obtain the image after style transformation. The VGG-19 network structure is depicted in Fig. 2.

As illustrated in Fig. 2, the network architecture of VGG-19 consists of 16 convolutional layers and three fully connected layers (*Sudha & Ganeshbabu, 2021*). VGGNet is not updated with weights during training and is solely utilized to calculate content loss and style loss at each level. Initially, the image starts as a noisy image X, and through continuous reduction of the loss function, the network is reverse-propagated. Ultimately,

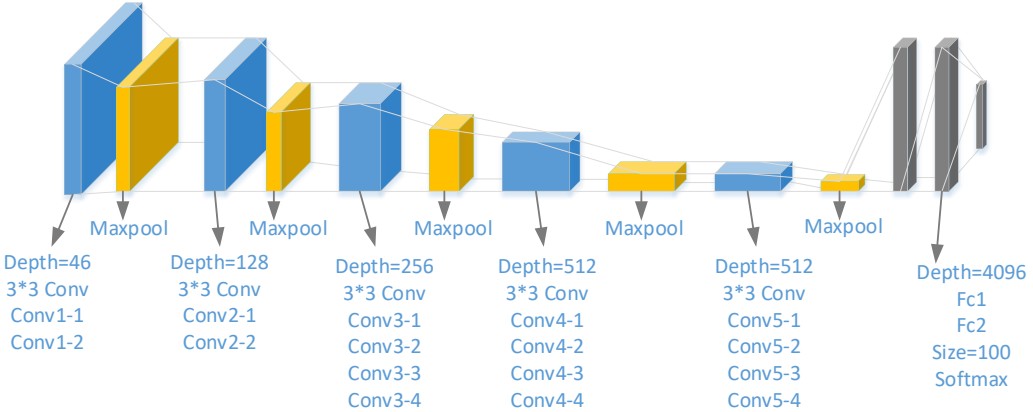

**Figure 2** VGG-19 network structure.

the noisy image X transforms into an image with the desired style. For this purpose, the implemented network structure must be compatible with VGGNet to compute the loss function effectively.

With increasing model depth, challenges such as gradient vanishing and exploding parameters emerged, impeding training efficacy and compromising model performance. Integrating the Inception module and residual connections proved pivotal in addressing these issues. The Inception module, characterized by its multi-scale receptive fields achieved through concurrent consideration of diverse convolutional kernel sizes, enhances the network's feature-capturing capabilities. Moreover, its incorporation of 1x1 convolutional kernels reduces channel dimensions, optimizing computational efficiency. Simultaneously, introducing residual connections, featuring skip connections that bypass particular layers and directly add input information to the output, facilitates smoother gradient flow during backpropagation, mitigating the gradient vanishing problem. Consequently, these innovations collectively augment the non-linear representative capacity of the VGG network, broaden the scope of feature extraction, and effectively alleviate gradient-related challenges, thereby enhancing the model's overall performance and training stability. This strategic amalgamation of the Inception module and residual connections renders deep convolutional neural networks more adept at intricate image recognition tasks.

Additionally, the model incorporates the use of residual connections. Within the Inception module, parallel convolution kernels of different scales extract rich feature information. The feature map information extracted from the pool3 layer is transmitted backward and input into the Concat layer, which merges with the feature map information from the conv4 layer based on the feature channel dimension. This fusion layer combines intricate details from the pool3 layer with comprehensive global feature information from the conv4 layer. This holistic approach maximizes the utilization of image-related data

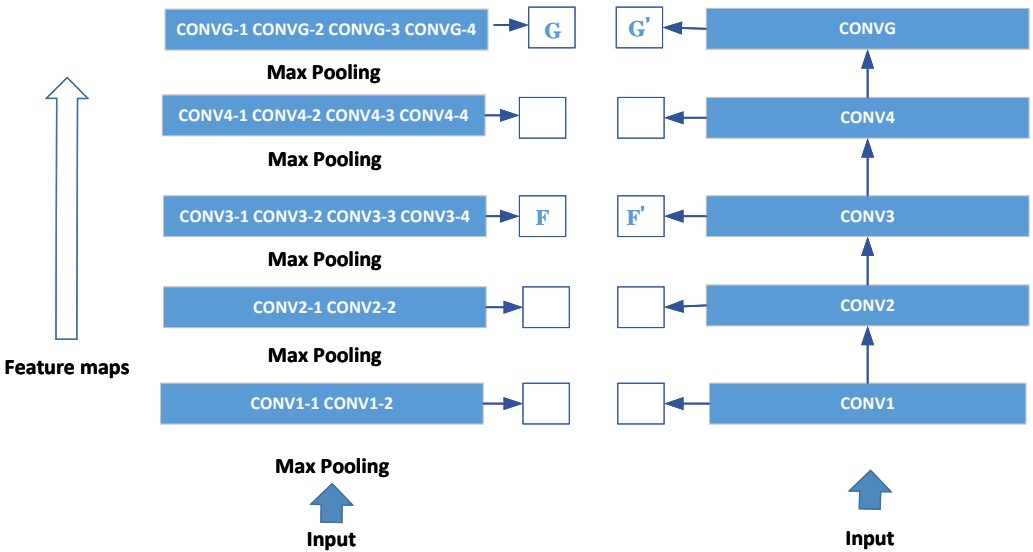

**Figure 3** Training process of the network in rendering style.

while mitigating the potential loss of essential image feature information. The training process of the network for rendering styles is depicted in Fig. 3.

Gram matrix is introduced for style definition, and the calculation process is shown in Eq. (1) (*Han & Li, 2023*).

$$G_{ij}^l = \sum_k F_{ik} F_{jk} \tag{1}$$

where $G_{ij}^l$ the inner product of the vector is composed of the i-th and j-th feature maps in layer l, and k represents the corresponding element of the feature map. Because the inner product of two vectors can determine the angle and azimuth relationship between vectors, it is often used to compare artistic painting styles.

The loss function of each layer style is defined as follows (*Monteiro, de Brito & Kurt Melcker, 2021*):

$$L_{\text{style}}^l(a,x) = \frac{1}{4N_l^2 M_l^2} \sum_{i,j} \left(G_{ij}^l - A_{ij}^l\right)^2 \tag{2}$$

where $N_l$ is the number of feature maps of this layer, and $M_l$ is the size of each feature map $A_{ij}^l$ represents the value of row i and column j of the style image $a$ of layer l. The primary function $\frac{1}{4N_l^2 M_l^2}$ is normalizing the order of magnitude of style and content loss.

In the actual programming process, multi-level networks are often selected to compare painting styles, and their weights are added up to get the final style loss (*Qiao-Lei, Song & Meng, 2018*). The specific calculation formula is:

$$L_{\text{style}}(a,x) = \sum w_l L_{\text{style}}^l(a,x) \tag{3}$$

where $w_l$ represents the style weight of each layer.

The final loss function is defined as follows:

$$L_{\text{total}}(p, a, x) = \alpha L_{\text{content}}(p, x) + \beta L_{\text{style}}(a, x) \qquad (4)$$

where, $a$ and $\beta$ represent the weight of each loss.

## Frontend implementation scheme

The frontend webpage of the rendering platform is built upon the latest version of Bootstrap. This architecture was introduced by the designers at Twitter Inc. in the United States. Bootstrap not only simplifies frontend development but also offers high efficiency and powerful performance, enabling the creation of an excellent webpage with minimal code. Furthermore, Bootstrap has undergone a complete redesign of most HTML controls, enhancing usability for mobile devices, and has evolved into an open-source project hosted on GitHub.

The primary function of the rendering platform's front page is to facilitate the upload of painting works to be rendered, the selection of traditional art styles for transformation, and the display of the rendered photos along with the rendering buttons.

## Back-end function realization

The server's back end on the rendering platform consists of three integral components. The first component is the URL matching request module, which routes URL requests to designated functions. The second component includes the logic processing module, which incorporates functions like management, conversion, encoding, and storage (*Lan, 2018*). The third component involves the rendering style algorithm, which converts user-uploaded artwork styles and subsequently delivers the transformed styles back to the users.

Figure 4 illustrates the interrelationships among these three modules within the background of the painting art and style rendering platform:

When the URL forwarding module receives a request from the front end, it initially assesses the legitimacy of the connection. If the request does not adhere to the defined rules, it promptly returns an error to alleviate pressure on the back end. In cases where the request is valid, it is matched with the corresponding function, and the required data is then transmitted to the function module. The presence of the URL forwarding module is crucial for enhancing the maintainability of the entire system. This module not only filters all frontend requests but also eases the workload on the system's logical function modules, thereby enhancing the system's scalability. With the forwarding module in place, when the system needs to add new functions, it can simply incorporate the corresponding function, and the forwarding module can transmit the requisite requests to this function without requiring modifications to the entire system.

In the back end of the rendering platform, several factors correspond to three distinct modules, each dedicated to specific functions. When a rendering request is sent by the front end and reaches the logic function module, the corresponding function initially extracts the information provided by the front end, primarily consisting of user-selected paintings and traditional artistic styles.

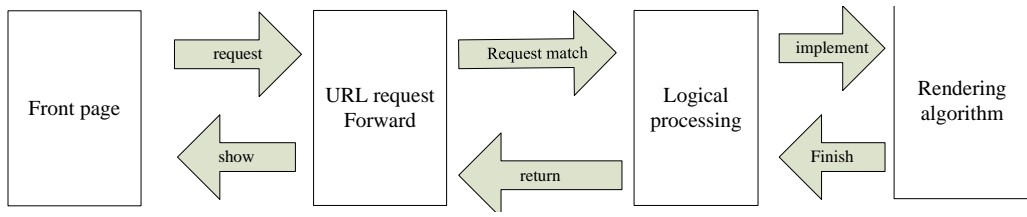

**Figure 4  Flowchart of the relationship between background modules.**

# EXPERIMENT AND ANALYSIS

## Experimental setup

The details of the experimental environment are shown in Table 1.

The training data is divided into two datasets, and for each training iteration, a set of data is randomly selected from these two datasets. Ultimately, network optimization and propagation are performed.

## Evaluation indicators

We apply the accuracy to evaluate the training performance of the VGG model, which can be expressed by:

$$Accuracy = \frac{TP + TN}{TP + TN + FP + FN}. \tag{5}$$

TP represents a valid example, TN represents a true counter-example, FP represents a false positive example, and FN represents a return example.

To conduct a more comprehensive assessment of the proposed embedded system's performance, we can enhance the evaluation by incorporating feedback times as a complementary metric.

Furthermore, we employ feedback time, which quantifies the number of cycles the system undergoes from receiving an input (*e.g.*, a user query) to producing an output (*e.g.*, an answer or recommendation result), to assess our system's performance. A shorter Feedback Time signifies that the system can respond swiftly, delivering more timely results and enhancing the user experience. We can evaluate the system's performance holistically by considering accuracy and feedback time. High accuracy ensures precise model classification, while a brief feedback time guarantees real-time efficiency, contributing to an optimal user experience. This combination of metrics provides a comprehensive understanding of the VGG model training and the proposed system's performance.

## Results

A total of one hundred natural images were used to conduct style transfer, employing both the original method and the method proposed in this article. This process resulted in two sets of one hundred stylized images each, totaling two hundred stylized images. Subsequently, a pre-trained VGG classification network was utilized to classify the natural images, the synthesized images generated through the original method, and those generated

**Table 1  Experiment parameters.**

| Component | Specification |
|---|---|
| Processor | Intel Core i7-12700 @ 2.10 GHz |
| RAM | 64 GB |
| GPU | NVIDIA Tesla V100 |
| Deep learning framework | PyTorch 1.11.0 |
| Operating system | Ubuntu 20.04 |
| Optimization method | Adam (momentum-based) |
| Initial learning rate | 0.002 |
| Batch size | 2 |
| Number of epochs | 1,000 |

*via* the proposed method. The recorded classification accuracy is visually represented in Fig. 5.

The VGG classification network exhibits relatively low accuracy when classifying natural images. Notably, the improved model demonstrates superior classification accuracy compared to three other models. Compared to the conventional CNN model, the enhanced model displays a 2.6% increase in classification accuracy, highlighting the advantage of incorporating the Inception module and residual connections for improving the model's classification capabilities. The improved model adeptly extracts multi-scale image information through the Inception module while mitigating the loss of crucial underlying details by applying residual connections; this, in turn, elevates classification accuracy, particularly in the context of Chinese painting images.

Furthermore, compared to the classical Lenet model, the improved model outperforms it by a substantial 12.15%. This significant enhancement primarily stems from the Inception module and residual connections, which augment the model's feature extraction capabilities. Additionally, incorporating batch normalization and overlapping pooling techniques in the improved model reduces overfitting.

Moreover, compared to the HOG + SVM algorithm, the improved model boasts a remarkable 15.98% increase in classification accuracy; this underscores the improved model's ability to effectively extract abstract features from Chinese painting images, surpassing the capabilities of traditional shallow learning algorithms and ultimately enhancing its classification proficiency. The classification accuracy of different image categories is shown in Fig. 6.

The provided figure illustrates that the method proposed in this article achieves the highest classification accuracy for architectural and landscape images, with rates of 87.34% and 95.33%, respectively. However, character classification exhibits a higher susceptibility to errors. This difference may be attributed to the straightforward differentiation of lines, textures, and frequency characteristics within architectural and landscape images. In contrast, synthesized stylized character images may contain noise artifacts, making it more challenging to incorporate feature information into the generator as prior knowledge accurately.

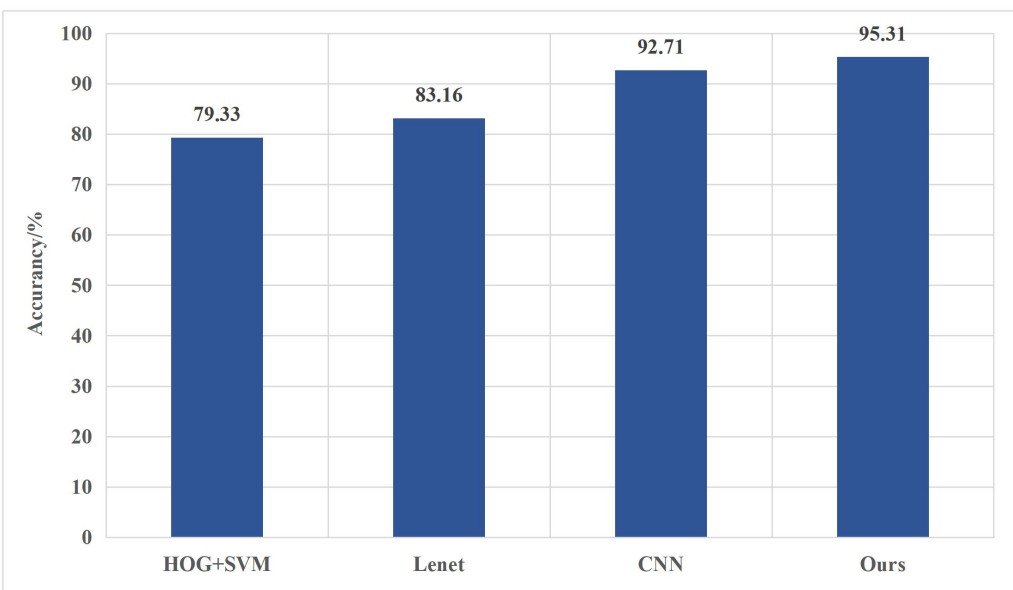

**Figure 5** Classification accuracy of different models.

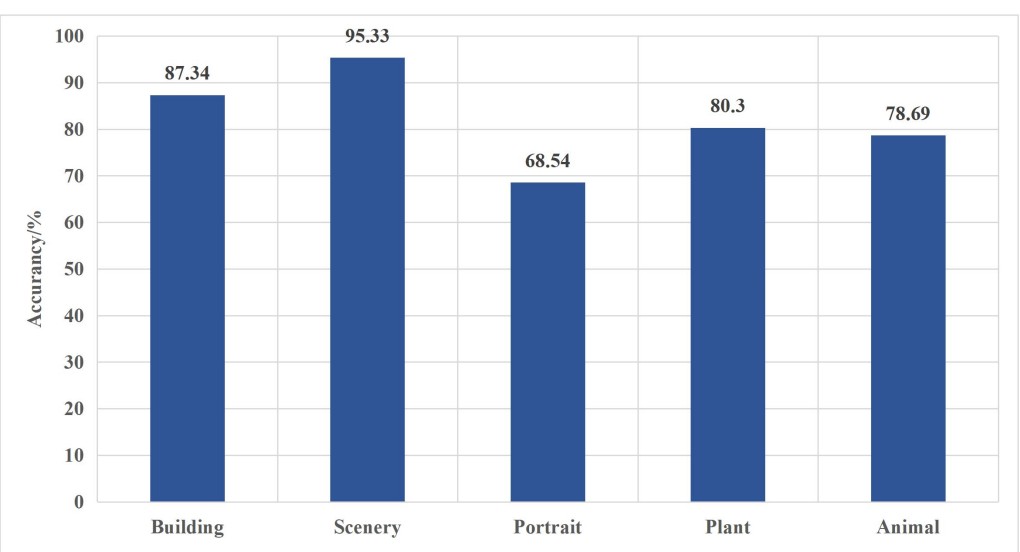

**Figure 6** Classification accuracy of different categories.

To further substantiate the system's impact on digital painting art and traditional art design, an assessment of its interactive performance was conducted, with results presented in Fig. 7.

The figure illustrates that as the number of image processing iterations increases, the system provides more detailed feedback on the style conversion process. This improved interaction between the operator and the system during the design process indicates

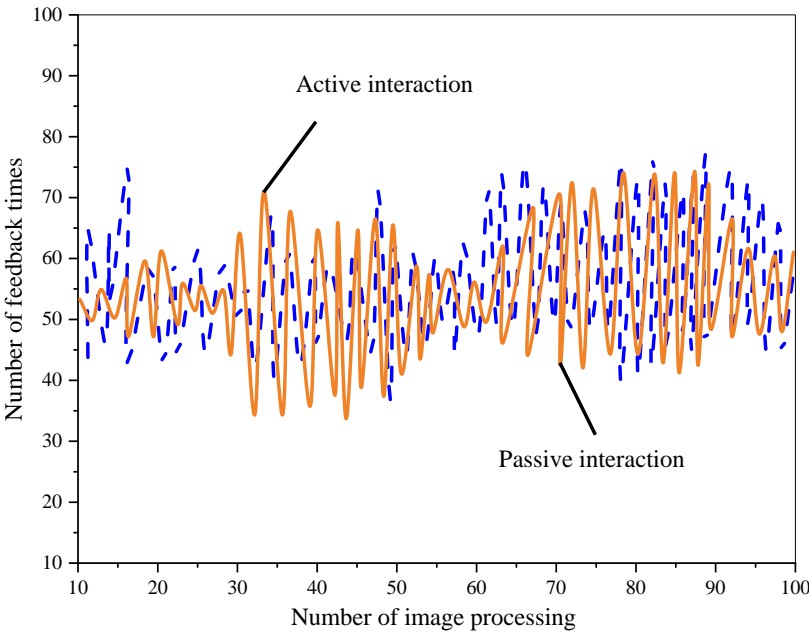

**Figure 7** **Test results of system interaction performance.**

a favorable level of collaboration and communication. Once the creative theme is set based on the reference style, the system will automatically perform the corresponding transformation according to the image content. Different depiction styles, composition methods, and color-matching choices will evoke varying emotional responses from the audience.

## DISCUSSION

Based on the above experiments, our method has demonstrated outstanding performance. The enhanced model, incorporating Inception modules, effectively captures multi-scale information from images and mitigates the loss of low-level details through residual connections, thereby strengthening the model's accuracy in classifying artistic photos. Additionally, integrating batch normalization and overlapping pooling techniques in the improved model reduces overfitting concerns, enabling it to outperform traditional shallow learning algorithms in extracting abstract features from artistic images and thus enhancing its classification capabilities. However, due to the limitations inherent in CNNs themselves, as the model's depth increases, it tends to focus more on semantic information, potentially overlooking some shallow texture and color details during classification, which can lead to inaccurate results.

The increase in image processing iterations directly corresponds to the system's feedback regarding style conversion intricacies. This phenomenon highlights a favorable and robust interaction between the operator and the design system. After configuring the creation theme and reference style as presets, the system performs the necessary transformations

based on image content, enabling seamless automatic processing. The diverse array of depiction styles, composition techniques, and color harmonization imparts unique emotional resonances to the audience, enriching their visual experience. From a system perspective, optimizing high-concurrency and multi-threaded design remains crucial to ensure the successful execution of image style conversion and effective utilization by multiple users.

Furthermore, this dynamic interplay between style conversion intricacies and feedback times underscores the system's adaptability and responsiveness, strengthening its potential as an efficient and intuitive design tool. Its inherent flexibility allows for creative exploration and experimentation, empowering operators on an enriched artistic journey. The system contributes to the symbiotic fusion of human ingenuity and machine intelligence by bridging the gap between human creativity and technological prowess. Moreover, the compelling amalgamation of different depiction styles, composition methods, and color palettes injects profound emotional depth into the visual narrative.

Despite the commendable achievements of the artistic image processing system described in this study, it still faces particular challenges. Achieving optimal stylized image outcomes requires careful manual parameter adjustments, especially when using model optimization methods, which often entail recurrent model re-training after each parameter modification. While a method for arbitrary style transfer without learning and training has been proposed to alleviate the issue of parameter manipulation and the need for separate model training for various styles, the complexity of the training process for this method hinders its effectiveness in image synthesis. Therefore, pursuing a more straightforward and controllable approach to ensure superior image quality becomes a central focus of future research efforts. Furthermore, while VGG serves as a substantial neural network model with a profound impact on image feature extraction, its computational overhead remains considerably high. As a result, the advancement toward micro-feature extractors emerges as a prospective direction for image style transfer rooted in neural networks.

Integrating the Bootstrap frontend framework with the Django web development framework represents a sophisticated approach to web application design, emphasizing a seamless interaction between the frontend and back-end components to optimize user experience. Bootstrap's responsive grid system facilitates the creation of adaptable layouts, ensuring a consistent and visually pleasing interface across diverse devices. This framework's pre-designed UI components and styles create a cohesive design aesthetic. Concurrently, Django's template system seamlessly incorporates Bootstrap's HTML and CSS components, enabling dynamic content generation utilizing Django's template language. Django's robust static file handling also ensures efficient delivery of Bootstrap assets, optimizing page load times. The amalgamation of these frameworks supports dynamic user interfaces by incorporating AJAX calls and integrating Bootstrap's JavaScript components, enhancing user interactions.

Furthermore, Django's authentication and authorization systems seamlessly integrate with Bootstrap's navigation components, facilitating user login/logout functionalities and role-based access control implementation. Regarding performance optimization, Django offers tools for caching and compressing content, while Bootstrap's styling complements

the overall aesthetic. Ultimately, the synthesis of Bootstrap and Django fosters the creation of contemporary, responsive web applications that prioritize functionality and user-centric design principles.

## CONCLUSION

Based on a comparative analysis of digital painting art and traditional art design, we propose a platform for rendering painting art styles based on an embedded system. The platform's front end is built on an embedded system that utilizes the Bootstrap framework, and Django serves as the back-end framework for web services. The front end primarily facilitates uploading photos for rendering, selecting styles for conversion, rendering through buttons, and displaying pictures after rendering. The platform's back end comprises three main modules: URL request forwarding, logic processing, and style rendering algorithms, all working in harmony to achieve the rendering and output of various painting art styles. This rendering platform, implemented on an embedded system, accomplishes the conversion and rendering of diverse painting styles using style-transferred networks and algorithms. The enhanced model effectively captures multi-scale information from images through the Inception module, mitigating the loss of essential details through residual connections. Consequently, it enhances the model's classification accuracy, mainly when applied to Chinese painting images. This advancement aims to enable personalized creation in digital painting art and traditional art design, fostering the creation of more exquisite digital works.

Furthermore, this study's style classification accuracy for portrait images could be higher. Future work will explore reducing noise artifacts in synthesized stylized images by incorporating total variation loss into the loss function. This approach aims to enhance the accuracy of style conversion further.

### Funding

The authors received no funding for this work.

### Competing Interests

The authors declare there are no competing interests.

### Author Contributions

- Yan Cui conceived and designed the experiments, performed the experiments, analyzed the data, authored or reviewed drafts of the article, and approved the final draft.
- Amer Shakir Bin Zainol conceived and designed the experiments, performed the computation work, prepared figures and/or tables, and approved the final draft.

### Data Availability

The code is available in the Supplementary File. The data is available at Zenodo: Europeana, & V4Design. (2021). V4Design/Europeana style dataset [Data set]. Zenodo. https://doi.org/10.5281/zenodo.4896487.

## Supplemental Information

Supplemental information for this article can be found online at http://dx.doi.org/10.7717/peerj-cs.2055#supplemental-information.

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
