# Peer review of "Traditional art design expression based on embedded system development"

_PeerJ Computer Science, doi:10.7717/peerj-cs.2055_

## Round 0.1 · original submission · Major Revisions

The reviews provide valuable insights into the proposed embedded system for a painting art style display platform, highlighting its strengths and improvement areas.

The first review emphasizes the need for further explanation regarding integrating Bootstrap and Django, incorporating TensorFlow architecture, and applying optimization techniques to the VGG network. Additionally, it suggests presenting experimental setup parameters in a tabular format for clarity.

The second review underscores the importance of contextualizing the improved image classification accuracy with related models, elaborating on the significance of specific accuracy peaks, and providing qualitative analysis to correlate style shifts with emotional impacts.

Please see both reviewers' comments. Integrating these suggestions would enhance the understanding and impact of the research findings.

**Language Note:** The review process has identified that the English language must be improved. PeerJ can provide language editing services - please contact us at [email protected] for pricing (be sure to provide your manuscript number and title). Alternatively, you should make your own arrangements to improve the language quality and provide details in your response letter. – PeerJ Staff

Reviewer 1 ·

Basic reporting

This paper describes the construction of the embedded system of painting art style display platform, which realizes the automatic integration of digital painting art and traditional art design. The experimental results show that compared with other models, the proposed model improves the accuracy of image style classification by 2.6%, and the accuracy of architectural and landscape image classification reaches 87.34% and 95.33% respectively. However, this paper still needs to improve the following:
Front-End Development with Bootstrap and Django Integration:

The utilization of the Bootstrap framework for front-end component design, coupled with Django as the web development framework, underscores a sophisticated approach to user interface design and web application development. This integration choice deserves additional elaboration in terms of how the frontend and backend seamlessly interact, especially in optimizing user experience.

TensorFlow Architecture Integration:
The incorporation of TensorFlow architecture into the code signifies a commitment to leveraging state-of-the-art machine learning tools. Providing more depth on how TensorFlow is integrated, the specific modules used, and the rationale behind choosing TensorFlow would enhance the technical understanding for readers and reviewers.

Enhancing VGG Network with Inception Module and Residual Connections:
The introduction of the Inception module and residual connections for optimizing the VGG network is a commendable strategy. To bolster the technical credibility of this choice, consider delving into the reasons behind selecting these specific techniques. Discuss how they contribute to enhancing the VGG network's performance, particularly in the context of texture feature recognition and analysis in images.

Experimental design

Experimental Methodology and Results Presentation:
Parameterization in Section 4.1:
Consider presenting the experimental setup parameters in a tabular format to enhance clarity and facilitate easy reference for readers and researchers. This will contribute to the overall professionalism and accessibility of the manuscript.

Validity of the findings

Comparison with Related Models:
While the experimental results showcase a noteworthy 2.6% increase in image style classification accuracy, a comparative analysis with related models is crucial for contextualizing this improvement. Explicitly reference and discuss other state-of-the-art models in the field, elucidating how your proposed model outperforms or complements existing approaches.

Class-Specific Accuracy Peaks:
The specific mention of achieving accuracy peaks of 87.34% and 95.33% in architectural and landscape image classification is compelling. Elaborate on the significance of these achievements and discuss any implications for real-world applications. Consider including visual representations, such as confusion matrices, to provide a more comprehensive understanding of the classification results.

Visualizing Style Shift in Image Processing:
In Figure 7, the representation of style shift as the number of iterations increases is intriguing. However, to enhance the connection with audience emotional response, consider incorporating a qualitative analysis. Perhaps include visual examples or perceptual studies that correlate the observed style shift with potential emotional impacts on viewers.

Novelty and Citation Enrichment:
Novelty of the Article: Clearly articulate the novel contributions of the article, emphasizing how the integration of Bootstrap, Django, TensorFlow, and the optimization techniques in the VGG network collectively advances the state of the art in image classification.

Additional comments

Citation Review:
Strengthen the scholarly foundation by revisiting and updating the citation list. Introduce references from reputable journals and recent publications that align with the scope of your research. This will fortify the article's academic standing and demonstrate a robust understanding of the current literature in the field.

By addressing these points, you can enrich the technical depth and completeness of the manuscript, ensuring that the reviewers and readers gain a comprehensive understanding of the innovative aspects and contributions of your research.

Cite this review as

Reviewer 2 ·

Basic reporting

Based on the comparative analysis of digital painting art and traditional art design, we propose a painting art style presentation platform based on embedded system. The operation results of the system show that the proposed platform reduces the burden of the system's logical function modules, enhances the scalability, and promotes the automatic integration of digital painting art and traditional art design expression.

Experimental design

Formula Explanation Improvement in Chapter 3:
 a. Comprehensive Explanation of Formulas (4) and (5):
 Formulas (4) and (5) in Chapter 3 lack a detailed explanation of their parameters. It is recommended that the author provide a comprehensive explanation of these formulas, elucidating the meaning and significance of each parameter to enhance the technical clarity for readers.

 Image Preprocessing Formula in Section 3.1:
 a. Expression of Image Preprocessing Formula:
 Although the image preprocessing steps are outlined in Section 3.1, the relevant formula expression is missing. Including the formula for image preprocessing will contribute to the technical rigor of the paper and enable readers to reproduce the preprocessing steps accurately.

Validity of the findings

Evaluation Indicators in Section 4.2:
 a. Completing Description of Evaluation Indicators:
 The description of evaluation indicators in Section 4.2 appears incomplete. Specifically, the feedback time mentioned lacks an introduction to the relevant formula. To address this, the author should provide a thorough explanation of the evaluation indicators and their associated formulas, ensuring a comprehensive understanding of the performance metrics.

 Enriching Experimental Results:
 a. Inclusion of Confusion Matrix:
 In the experimental results section, it is advisable to enhance the richness of the experiment by incorporating additional visual aids, such as confusion matrices. This would provide a more detailed and nuanced perspective on the model's performance across different classes.

Additional comments

Streamlining the Discussion Section:
 a. Focus on Key Points:
 The discussion section is noted to be excessively lengthy. To optimize its impact, focus on key points related to research content, introduction, and future prospects. This will streamline the discussion and make it more impactful and reader-friendly.

 Professional Language Usage:
 a. Re-polishing Language for Professionalism:
 The overall language in various parts of the article is observed to lack professionalism. Consider revising and polishing the language to align with the expected standards of scientific discourse. This includes using precise technical terminology and ensuring clarity in expression for a more refined and professional presentation

Cite this review as

---

## Round 0.2 · accepted · Accept

Both reviewers confirmed that the authors had addressed all of their comments.

Reviewer 1 ·

Basic reporting

Satisfactory

Experimental design

Good

Validity of the findings

Good

Cite this review as

Reviewer 2 ·

Basic reporting

The paper presents a pioneering approach to constructing an embedded system that seamlessly integrates digital painting art with traditional art design. This innovative platform bridges the gap between modern digital techniques and traditional artistic expression, opening new avenues for artistic exploration and creativity.

Experimental design

The experimental results demonstrate the effectiveness of the proposed model, showcasing a notable increase in image style classification accuracy. With a 2.6% improvement, the proposed model achieves impressive accuracy rates of 87.34% and 95.33% in architectural and landscape image classification, respectively

Validity of the findings

outcomes of the platform highlight its practical benefits, including alleviating the burden on logical function modules, enhancing scalability, and promoting the automated fusion of digital painting art with traditional art design expression

Additional comments

The revised paper seems okay to proceed

Cite this review as